# Utilization of Finite Element Analysis for Articular Cartilage Tissue Engineering

**DOI:** 10.3390/ma12203331

**Published:** 2019-10-12

**Authors:** Chaudhry R. Hassan, Yi-Xian Qin, David E. Komatsu, Sardar M.Z. Uddin

**Affiliations:** 1Department of Biomedical Engineering, Stony Brook University, Stony Brook, NY 11794, USA; Chaudhry.Hassan@stonybrook.edu (C.R.H.); Yi-Xian.Qin@stonybrook.edu (Y.-X.Q.); 2Department of Orthopaedics, Stony Brook University, Stony Brook, NY 11794, USA; david.komatsu@stonybrookmedicine.edu

**Keywords:** articular cartilage, tissue engineering, scaffold design, finite element analysis

## Abstract

Scaffold design plays an essential role in tissue engineering of articular cartilage by providing the appropriate mechanical and biological environment for chondrocytes to proliferate and function. Optimization of scaffold design to generate tissue-engineered cartilage has traditionally been conducted using in-vitro and in-vivo models. Recent advances in computational analysis allow us to significantly decrease the time and cost of scaffold optimization using finite element analysis (FEA). FEA is an in-silico analysis technique that allows for scaffold design optimization by predicting mechanical responses of cells and scaffolds under applied loads. Finite element analyses can potentially mimic the morphology of cartilage using mesh elements (tetrahedral, hexahedral), material properties (elastic, hyperelastic, poroelastic, composite), physiological loads by applying loading conditions (static, dynamic), and constitutive stress–strain equations (linear, porous–elastic, biphasic). Furthermore, FEA can be applied to the study of the effects of dynamic loading, material properties cell differentiation, cell activity, scaffold structure optimization, and interstitial fluid flow, in isolated or combined multi-scale models. This review covers recent studies and trends in the use of FEA for cartilage tissue engineering and scaffold design.

## 1. Introduction

Articular cartilage is predominantly made up of chondrocytes that are differentiated from mesenchymal stem cells (MSCs) [1]. The spatial orientation of cartilage is defined by the organization of chondrocytes and the extracellular matrix in three distinct layers [2]. The upper superficial layer contains flattened layers of chondrocytes with collagen fibers oriented parallel to the articular surface. The middle layer contains oblique chondrocytes with a random orientation of collagen fibers. Finally, in the deep layer close to the bone, chondrocytes are oriented radially with a perpendicular collagen fiber orientation [3,4]. Cellular morphology and extracellular orientation are both regulated by mechanical stimuli [5,6,7]. Mechanical stimuli induce conformational changes in integrins, thereby regulating gene expression and tissue remodeling through the process of mechanotransduction [8]. Chondrogenic mechanical stimuli can comprise compressive or shear forces that are dependent on amplitude, direction, and frequency [9,10]. Proper mechanical stimuli are vital to cartilage homeostasis, as well as regeneration. Importantly, lack of mechanical stimulus, along with aging, inflammation, and obesity, are risk factors for the development of osteoarthritis (OA) [11]. Despite the fact that 30 million adults are currently diagnosed with OA in the US, there are no good treatments for this disease, and the degeneration of articular cartilage resulting from OA, as well as other cartilage disorders, would greatly benefit from functional tissue-engineered cartilage [12]. Scaffolds have the potential to provide the proper mechanical and spatial environment for chondrocytes to proliferate and generate functional tissue-engineered cartilage in order to meet this demand.

Scaffold design is a critical process in the engineering of functional cartilage that can ensure appropriate interactions between the cells and the scaffold [13]. The design process requires sequential in-vitro, mechanical, and in-vivo tests to determine the optimal structural parameters for the desired level of mechanotransduction [14]. Conventionally, designing a scaffold has been based on a “trial and error” approach: Incremental modifications of previous designs are carried out to determine a new design [13]. As the optimization of scaffolds for clinical applications needs to be tested extensively using in-vitro and in-vivo systems, this has been a time-consuming process. To overcome these limitations in scaffold optimization, finite element analysis (FEA) has gained popularity over the years as a preliminary in-silico step for scaffold design.

FEA is a computational mechanics tool that performs stress–strain analysis within a body (scaffold) by dividing it into smaller blocks (elements) of an approximately regular shape. These shapes can be 2D (planer triangle or quadrilateral) or 3D (tetrahedral or hexahedral) and are formed by placing nodes on the solid geometry. The most basic 3D element shape is a tetrahedron made up of four nodes. A combination of tetrahedrons can create an eight-node hexahedron (Figure 1) [15]. Advanced models use higher-order 20-node hexahedral elements, thereby providing more accurate analyses. A mathematical constitutive equation is then applied and solved for the stress–strain at each node. The analysis can use simple linear elastic [16] or complex biphasic elastic formulations [17,18]. Linear elastic material constitutive equations assume infinitesimal strains and obey Hooke’s Law (stress is linearly proportional to strain) [16]. In contrast, biphasic material analysis is a solid-fluid coupled stress–strain formulation, where the solution is dependent on elastic modulus, Poisson’s ratio (bulk modulus), and permeability of the matrix [19]. FEA provides the ability to predict structural deformation, stress distribution, and cartilage tissue regeneration within composite scaffold structures [14,20]. The availability of high-end processors for laboratory use has enabled researchers to design and analyze scaffolds in silico with a range of loading and boundary conditions, non-invasively and rapidly [21]. Finer mesh models with higher numbers of nodes increase the capability of predicting cellular responses to mechanical loading [22]. These cellular models identify the optimal design and mechanical stimuli needed for mechanotransduction [23]. In this paper, we will review recent advancements in FEA for cartilage scaffold design, optimization, and the impact this has on improving scaffold design.

## 2. Components of Finite Element Analysis

### 2.1. Development of Scaffold Models

The first step in FEA is generating the mesh from solid geometry or radiology scans. Mesh with four-node tetrahedrons is the most basic type of 3D mesh. The scaffold structure in tetrahedral mesh models usually does not have fibers, and hence, computationally expensive biphasic material formulations can be solved with a linear tetrahedral mesh to save FEA time [24]. Figure 2A shows a scaffold structure consisting of a grid-like scaffold with embedded cells. Figure 2B shows a basic model of a tetrahedral mesh cell with 6000 nodes on the surface, which is not ideal for cartilage scaffolds and the cartilage matrix [25]. Fibrous structures are usually modeled as composites, with the matrix modeled as hexahedral element mesh and fibers embedded inside. Eight-node hexahedral brick-shape elements are commonly used for the matrix and two-node 2D truss elements for the fibers [17,26,27]. Three-dimensional hexahedral element mesh without fibers has been used to represent scaffolds. [28]. A higher-order 20-node formulation of hexahedral elements has also been used to represent tri-quadratic interpolation of displacements [29]. These higher-order node-based elements have higher accuracy but come with a higher computation cost [30].

### 2.2. Constitutive Equations

Native human articular cartilage is a fluid-filled tissue where the fluid swelling pressure provides resistance to the compressive loading. To mimic the physiological environment of cartilage, computational models should include solid and fluid phases with an assigned fluid volume ratio or a void ratio (i.e., porosity) and permeability. Multiple groups have demonstrated the utility of biphasic solid-fluid and poroelastic cartilage scaffolds models [17,26,28,31]. The matrix is defined either as a linear elastic [27,31] or non-linear elastic isotropic material [26]. Linear elastic models were assigned Young’s modulus (E) and Poisson’s ratio (v) to the solid phase of the scaffold of: E = 34.99 kPa, v = 0.495 [27]; E = 133 kPa, v = 0.3 [31]; and E = 9.5 kPa, v = 0.3 [17]. Poro-elastic and biphasic models are assigned porosity, fluid permeability, solid bulk modulus, and fluid bulk modulus to simulate fluid-filled tissues. The porosity value of a scaffold varies for each design and is usually input as void ratio in the equation. Void ratios of three (75% porous) [31] and four (80% porous) [28,32] have been assigned to scaffolds, though other values based on different designs can be found in the literature. Fluid permeability is either defined as a uniform value or deformation-dependent exponential function based on van der Voet’ formulation with initial permeability assigned to the solid matrix [31]. Permeability values of 5 × 10^−12^ m^4^ N^–1^ s^–1^ [28], 60 × 10^–12^ m^4^ N^–1^ s^–1^ [17], and 1 × 10^−14^ m^4^ N^–1^ s^–1^ [32] have been assigned to different kinds of scaffolds. The pores consist of connected channels in a solid structure filled with interstitial fluid [26]. Cartilage is a non-linear elastic material; therefore, some FE models have used customized user material subroutines (UMAT) to incorporate hyperelastic, porous solid matrices [33]. Hyperelastic material properties allow incorporation of non-linear stress–strain responses in the scaffold and engineered cartilage modeling; Neo-Hookean [22,26,32], Ogden [22] and Yeoh [34] constitutive equations have been used to model the hyperelastic properties of tissue-engineered scaffolds and cartilage.

Representation of the fibers in the constitutive model of the matrix is important for the analysis of native cartilage or scaffolds. Fibers are usually modeled as isotropic materials, with either linear or non-linear (hyperelastic) constitutive equations. Ahn et al. modeled poly (lactic-co-glycolic acid) (PLGA) fibers as transverse isotropic materials with different moduli in axial and transverse directions for their scaffold [35]. FEA predicted that deformations in the scaffold structure were then validated with the fabricated scaffold under compressive and tensile loads. Cortez et al. modeled hydrogel polycaprolactone (PCL) fibers as an anisotropic material based on Holzapfel’s constitutive equation [17]. Their results concluded that the fibers realigned themselves differently in each zone of the construct to resist the applied load. Anisotropic representation of the fibers is more realistic as the fiber distribution in articular cartilage changes between zones [36]. Therefore, any FEA of scaffold design should include orientation and depth-dependent fiber distribution. Moreover, the FEA of scaffolds based on hyperelastic matrices and depth-dependent fibers is essential for accurate prediction of mechanical responses.

The mechanical parameters input in the constitutive equations are determined through mechanical tests. Cartilage and fluid-filled soft materials exhibit a non-linear stress–strain response. An unconfined compression test provides compressive modulus of the sample in wet or fluid submerged conditions [27,32]. Compression test protocols include stress–relaxation where a strain of up to 20% is applied as a ramp and held at a constant load for several minutes to allow the stresses to reach equilibrium. The ramp region is used to calculate the compressive modulus, and the relaxation curve is used to calculate the equilibrium modulus. Fiber mechanical properties are determined through a uniaxial tensile test under a rate-dependent strain [27]. The force-displacement data are curve-fitted to obtain hyperelastic equation coefficients.

### 2.3. Loading Conditions

FEA has the advantage of testing multiple loading modalities to extensively study the performance of scaffolds. These loading conditions serve as an input to the chosen constitutive equations, and scaffold responses are determined based on the corrosponding solution. The most common way to test scaffolds is under static forces [14,37] and strains [27,38,39]. A common use of a static load is to perform parametric analysis with a range of static forces in order to determine the force required to generate physiologically appropriate strains in a scaffold or cartilage tissue [40]. Loading rates are also important as cartilage is a viscous tissue which exhibits time-dependent stress responses under dynamic forces and displacements. Similarly, scaffolds with tissue ingrowth require stress relaxation responses to validate the structural integrity of the newly formed tissue. Meloni et al. simulated unconfined stress relaxation from 10% strain to determine the force-time response of a cartilage subchondral bone construct [26]. In a poroelastic model with fluid inside a solid extracellular matrix or scaffold, stress relaxation and indentation tests provide long-term force profiles of the materials.

Although static loads provide the stress profile for scaffolds, they are insufficient to test the mechanical strength of cartilage or scaffolds. Cartilage experiences diverse in-vivo loads based on activity, like walking, running, and jumping induce different dynamic loads applied at different rates and come in different frequencies. Dynamic loads have been commonly used to validate scaffold designs [17,28,41]. Bandeiras et al. combined dynamic strain in compression, bending, and shear to determine the mechanical stimulation parameters required for solute supply and Young’s modulus remodeling [28]. Similarly, Cortez et al. applied physiological strains of 15% at 1 Hz in a biphasic scaffold model to predict collagen fiber realignment [17]. Their results showed a depth-dependent fiber realignment under dynamic loading. Low-frequency dynamic loading best replicates walking or running conditions and, therefore, provides the mechanical parameters necessary to determine a scaffold’s strength during daily activity.

### 2.4. Finite Element Software

FEA requires software tools for mesh development, pre-processing, solving, and post-processing stages. Each FEA stage has specialized commercial and freeware software programs. Some of the mesh development and pro-processing softwares include Mimics (Materialise, MI, USA) [42], Hypermesh (Altair Engineering, MI, USA) [43], Trelis (version 15.1.3, csimsoft^TM^, UT, USA) [26], CATIA (Dassault Systemes, RI, USA) [31], and FEBio (version 1.8, Salt Lake City, UT, USA) [26]. Mimics specialize in surface extraction from image data and 3D mesh generation on the surfaces. FEBio allows pre-processing of experimental mechanical test data to generate parameters such as hyperelastic coefficients. Abaqus (versions 6.5, 6.10, 6.12, Dassault Systemes, RI, USA) [14,28,31,44] is the most used FE solver in research due to its strength in structural mechanics computational analysis, non-linear contact solving, and availability of porous medium material library. Other solvers used in the reviwed scaffold FEA include Ansys (PA, USA) [45], COMSOL (version 3.5, Burlington, MA, USA) [46], and FEBio [26]. These solvers also have a visualization module that allows for post-processing of simulated results.

## 3. Applications of Finite Element Analysis

The use of FEA in cartilage tissue engineering encompasses cell to tissue level biomechanics. Several studies have developed mesh models of the scaffolds, cartilage, and cells, as well as bioreactors to predict the mechanical parameters (stress and strain) under different load profiles. In the following section, we describe the application of some of these models in cartilage tissue engineering and scaffold design.

### 3.1. Cellular Activity

Cellular-scale FE models have been developed to determine cell viability by predicting the biophysical and biomechanical responses of scaffolds. Byrne et al. simulated cell migration and proliferation in a multi-scale model of scaffold, granulated tissue, and cells (Figure 3) [47]. Their model allowed lattice cells to proliferate in three dimensions based on a random walk algorithm. Depending on cell type, the model allowed either proliferation or migration during the simulation. Elsaadany et al. developed a multi-length scale FE model to predict regional strain gradients in a cell-embedded collagen construct [22]. The model used in their study had a macro component focusing on the mechanics of the collagen construct and a micro-component predicting the cellular responses to the strains. Their model demonstrated its ability to predict applied strains being transferred from the collagen scaffold to the embedded cells for cell viability analysis. Riccardo et al. used a similar approach to model a bioreactor chamber and predicted the strain effects of a macro-scale scaffold on biomass growth at the micro-scale of the cellular construct [48]. Similar models have been developed by other researchers to predict cellular biomechanical responses and viability under varying magnitudes of applied strains [46], as well as cell differentiation based on fluid shear stress [49]. These models with cell–matrix interfaces can be used to optimize scaffold design and applied stress and strain magnitudes to induce suitable cellular responses [46]. Sandino and Lacroix simulated perfusion fluid shear stress as a mechanical stimulus instead of fluid velocity, to stimulate tissue growth in a porous scaffold [49]. Their model predicted variation in the fluid shear stresses as the cells seeded in the scaffold formed new tissue. These published multi-scale models have demonstrated the ability to predict cell activity under applied strains and shear stress loads on the scaffold structure and can contribute to scaffold design by accurately predicting the cellular responses inside these changing structures [22,49].

### 3.2. Cell Differentiation

The differentiation potential of cells in scaffolds has been computationally analyzed to determine the optimal parameters for tissue growth [23,50]. Xue et al. simulated a co-culture bioreactor to predict chondrogenic and osteogenic differentiation for cartilage and subchondral bone, respectively [23]. Their methodology involved a bioreactor with two inlets separately for the chondrogenic and osteogenic components of the scaffold. The scaffold was placed in a cylinder in the middle of the bioreactor, and perfusion fluid was allowed to flow from the inlets through the scaffold to the outlet. Fluid flow velocity, fluid-induced shear stress, and media mixing were predicted through FEA and were found to support osteochondral differentiation. Sandino et al. studied tissue differentiation in the pores of calcium phosphate scaffolds with irregular morphology and predicted that mechanical compressive strain and angiogenesis would influence tissue differentiation within the scaffold [50]. Scaffold mechanical properties and geometry were also predicted to influence cell differentiation [51]. These models demonstrate the ability of FEA to predict cell differentiation under different designs and loading conditions; thus, FEA can be used to design scaffolds for appropriate tissue differentiation and growth.

### 3.3. Scaffold Structural Optimization

Optimization of scaffold design and mechanical properties is not only cheaper but also more mechanically extensive in silico. Mechanical properties of scaffolds require optimization to maximize cartilage tissue formation after the differentiation of MSCs [14]. Koh et al. used a two-model approach for this goal: a poroelastic model and a cellular model, with the sharing of strain values between the two models [14]. Their model allowed site-specific loading and optimized material properties for the superficial, middle, and deep zones of the scaffold. Another study optimized porosity of a bone scaffold under compressive, shear, and combined compressive-shear forces and predicted that gradient changes in the scaffold porosity would yield more tissue than a scaffold with homogenous porosity [52]. Olivares et al. performed a parametric analysis of six scaffolds with diverse porosity and designed to optimize fluid flow and axial compressive strain for tissue differentiation (Figure 4) [24]. Their analysis predicted that gyroid pore structures are better for cell attachment, due to a larger surface area under the influence of mechanical stimuli. Kelly and Prendergast performed a parametric study to predict the optimal scaffold mechanical properties to maximize the amount of cartilage tissue and reduce the amount of undesired fibrous tissue [53]. Their analysis predicted that increasing the stiffness of scaffold up to a certain threshold and decreasing the permeability of the scaffold would yield the maximum cartilage tissue. These studies show that optimization of scaffold microstructure to reach functional mechanical and mass transport environments can be achieved by parametric analysis of bulk modulus, diffusivity, and porosity [54].

### 3.4. Fluid Flow

Cartilage tissue regeneration in a scaffold is dependent on fluid flow for mechanical stimulus [55] and fluid pressure [56]. Porous medium computational analysis of scaffolds and cartilage allows for a prediction of fluid velocity and fluid pressure distribution in the fluid phase. Mesallati et al. studied the effects of dynamic compression-induced fluid flow on the glycosaminoglycan quantity in scaffolds [29]. Their model predicted the highest fluid flow at the solid hydrogel scaffold boundary, whereas the heterogeneous distribution of fluid flow and fluid pressure was predicted in channel-based scaffolds, both under dynamic compression. They concluded that a channeled scaffold provides a larger effect on fluid flex and, therefore, can promote collagen synthesis through fluid-induced shear stress. Chen et al. used in-silico biphasic FEA of poly (vinyl alcohol) (PVA)-cartilage constructs to characterize interface micromotion, stress, and fluid flow under cyclic loading of 6–40 N at 1 Hz. The model predicted a decreased scaffold–cartilage interface under cyclic loading, and this prediction was verified using scaffold and cartilage explants that mimicked the bone–cartilage interface [57].

## 4. Summary and Future Directions

Cartilage tissue engineering and scaffold design have been positively impacted by the use of FEA. FE models provide iterative design and analysis capability for the tissue engineering researchers, as they can be used to mechanically and biologically optimize scaffolds. Loading conditions in scaffold models can be varied to determine the appropriate type and magnitude of loading. Advancements in computational power and ever-maturing FE models provide unique tools to develop and test new hypotheses for cartilage tissue engineering. The ability of these models to accurately predict the stress, strain, fluid pressure, flow, and cell growth enables rigorous in-silico analyses of scaffold designs before they are actually produced.

A remaining challenge in tissue-engineering functional cartilage is the scarcity of zone-dependent scaffold models. There is also a need for studies that predict micro-motion at the cartilage-scaffold interface, as well as those that can analyze structural discontinuity between the engineered tissue and native cartilage. In this aspect, the approach used in ligament tissue engineering can be implemented for cartilage, as both of these tissues are non-linear elastic and contain direction-dependent fibers [58]. Fiber remodeling under changing mechanical and biochemical environments warrants in-silico implementation. Preliminary theory and algorithms on cartilage fiber remodeling and orientation have been developed [59,60] and require implementation in FE models in the future. Lastly, as the complexity of models keeps increasing and we move to multi-scale models, computational resources and the cost necessary to acquire the latest systems may become a hindrance for smaller laboratories. Therefore, sharing and optimization of FE models and software routines is important to minimize costs and maximize the impact of FEA on cartilage tissue engineering.

## Figures and Tables

**Figure 1 materials-12-03331-f001:**
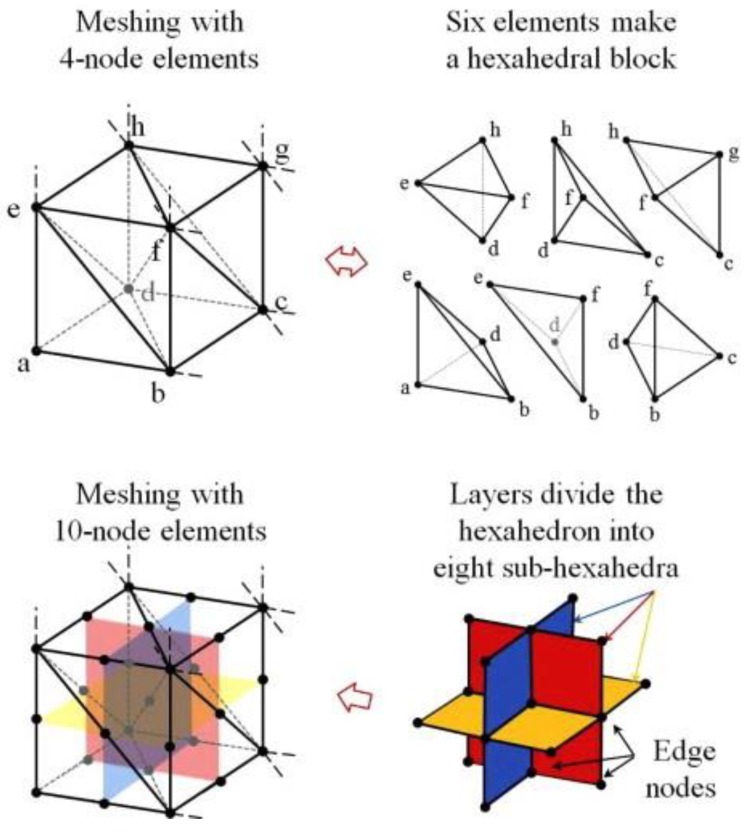
Tetrahedral (four-node) and hexahedral (eight-node and 20-node) elements are commonly used in the finite element mesh. The solid body is replaced with the selected type of elements and the stress–strain constitutive equations are solved at each node. (Figure taken from Kim, J., & Bathe, K. J. [15] and reprinted with permission).

**Figure 2 materials-12-03331-f002:**
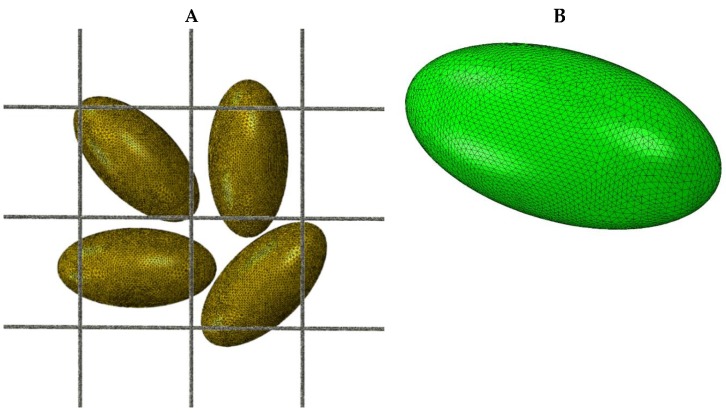
(**A**) A representative model of a scaffold with embedded cells. The cell surface nodes are in contact with the scaffold’s rod-like structure. The force applied to the scaffold results in strain on cell surface nodes. (**B**) A detailed tetrahedral mesh of the ellipsoid shaped cell with a 100-µm major axis and a 50-µm minor axis that contains 6000 nodes.

**Figure 3 materials-12-03331-f003:**
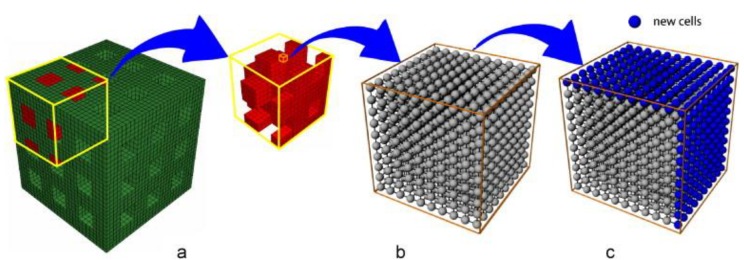
(**a**) Finite element model of a scaffold with 50% porosity. The scaffold cavity is initially filled with granulation tissue (red). (**b**) Each element of the granulation tissue has a lattice for analysis of cellular activity. (**c**) The lattice expands by the addition of new cells (blue) to grow into the space of the dissolving scaffold material. (Figure is taken from Byrne et al. [47] and reprinted with permission).

**Figure 4 materials-12-03331-f004:**
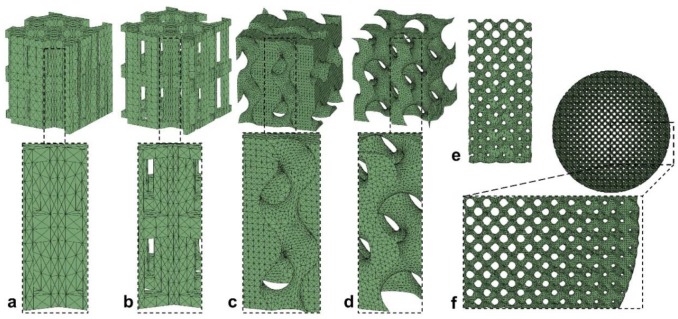
Six scaffold designs with varying structure and porosity; hexagonal prisms with (**a**) 55% and (**b**) 70% porosity, gyroid with (**c**) 55% and (**d**) 70% porosity, (**e**) gyroid with height-based porosity gradient, and (**f**) gyroid with radial-based porosity gradient. (Figure is taken from Olivares et al. [24] and reprinted with permission).

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
