# Peer review of "Utilization of Finite Element Analysis for Articular Cartilage Tissue Engineering"

_materials, 2019, doi:10.3390/ma12203331_

Round 1

Reviewer 1 Report

The aim of this manuscript is to review the recent papers published in the topic of Finite Element Simulation used in the cartilage tissue engineering and design of scaffolds.

Overall, this paper summarizes well the current state of knowledge of the topic and creates an understanding of the topic for the readers.

In my opinion, it would be interesting for readers to include a chapter or subchapter containing discussions about the FEA software and associated software (such as preprocessing software) suitable for performing FEA in this topic, as well as about the material parameters that are used as input data in the constitutive models to describe the behavior of materials during the simulation. Furthermore, it would be interesting to discuss the experimental tests necessary to determine the mechanical parameters of materials used as input data in the Finite Element codes; for example, how are bone and cartilage material properties determined?

These concerns may be equally critical in the use of FEA in the cartilage tissue engineering and design of scaffolds.

Author Response

The aim of this manuscript is to review the recent papers published in the topic of Finite Element Simulation used in the cartilage tissue engineering and design of scaffolds.

Overall, this paper summarizes well the current state of knowledge of the topic and creates an understanding of the topic for the readers.

In my opinion, it would be interesting for readers to include a chapter or subchapter containing discussions about the FEA software and associated software (such as preprocessing software) suitable for performing FEA in this topic, as well as about the material parameters that are used as input data in the constitutive models to describe the behavior of materials during the simulation.

A new section has been added in the manuscript, titled “2.4. Finite Element Software” which lists the finite element analysis pre-processing and post-processing softwares used by researchers in the field of cartilage scaffold FEA.

Furthermore, it would be interesting to discuss the experimental tests necessary to determine the mechanical parameters of materials used as input data in the Finite Element codes; for example, how are bone and cartilage material properties determined?

Section 2.2. “Constitutive Equations” has been revised to include a paragraph on two commonly used tests to determine material properties tissues: compression tests with stress relaxation, and tensile tests.

These concerns may be equally critical in the use of FEA in cartilage tissue engineering and the design of scaffolds.

Reviewer 2 Report

The paper provides a good overview of the achievements in the field of numerical modeling of cartilage tissue. In the whole work, I miss more accurate information on the conditions of FEM simulation (input data for the simulation, material parameters such as Young's modulus, Poisson's ratio, etc.), which would help in the overall assessment of the work. Nevertheless, the work is worth publishing because it has cognitive values.

FEM analysis is a very useful research tool at the moment but requires very precise selection of simulation parameters (eg. DOI: 10.12913/22998624/64064)

Author Response

The paper provides a good overview of the achievements in the field of numerical modeling of cartilage tissue. In the whole work, I miss more accurate information on the conditions of FEM simulation (input data for the simulation, material parameters such as Young's modulus, Poisson's ratio, etc.), which would help in the overall assessment of the work.

Section 2.2. “Constitutive Equations” has been revised to include cartilage scaffold material parameters such as Young’s modulus, Poisson’s ratio, permeability, and void ratio, as used in the cited literature.

Section 2.3. “Loading conditions” provide the input load and boundary conditions during the model set up and before the finite element solver is run.

Nevertheless, the work is worth publishing because it has cognitive values.

FEM analysis is a very useful research tool at the moment but requires very precise selection of simulation parameters (eg. DOI: 10.12913/22998624/64064)

A few more papers, including the one mentioned by the reviewer, have been included in the manuscript to elaborate on simulation parameters.